# “Not Thinking that This Means the End When You Are Seriously Ill but Doing Something Positive about It”—A Qualitative Study of Living with A Life-Changing Disease

**DOI:** 10.3390/diseases7030053

**Published:** 2019-09-11

**Authors:** Inger Benkel, Elin Ljungqvist, Maria Arnby, Ulla Molander

**Affiliations:** 1Sahlgrenska Academy, Institute of Medicine, Geriatric Medicine and Clinical Osteoporosis Research School, University of Gothenburg, 4405 30 Gothenburg, Sweden; ulla.molander@vgregion.se; 2Palliative Care Unit, Sahlgrenska University Hospital, 400 43 Gothenburg, Sweden; maria.arnby@vgregion.se; 3Regional Cancer Centre West, Sahlgrenska University Hospital, 413 45 Gothenburg, Sweden; elin.ljungqvist@rccvast.se

**Keywords:** chronic disease, turning point, strategies, support, life-changing

## Abstract

Background: Chronic diseases have an impact on and can change the lives of the persons affected by them. This study examines how a disease can influence patients’ daily lives, the strategies they adopt to cope, and their experiences of support. The study focuses on four chronic diseases: asthma-allergy, cancer, diabetes mellitus, and inflammatory rheumatic arthritis. Methods: The study has a qualitative design and includes 41 transcribed in-depth interviews and a content analysis. Results: The participants’ new life situation was changed for a very long time or forever, and this was not a voluntary choice. The new life situation comprised the following themes: life-changing—the disease could be a turning point in a negative or positive way, strategies—designed to create ways of coping with daily tasks to find a good quality of life, and support—that could be obtained from the participants’ private network or the healthcare professionals. Conclusions: The patients had to make changes in their daily life, and these could bring about different feelings and restrict activity. Healthcare professionals need more knowledge of the process of coping with such life-changing matters and what could strengthen patients and give a sense of empowerment in their lives.

## 1. Introduction

A chronic disease takes people on an inevitable journey comprising a profound process of intrapersonal changes, and can lead to an improved or worsened daily life. A life-changing disease is understood as an unpredictable situation in which a person is faced with particular demands caused by a chronic and potentially life-threatening condition [1,2]. Chronic diseases have an impact on and change their daily lives for the people affected by them for a long time; they can also be life-threatening [3]. Information about the disease and rehabilitation are important to help patients cope with their new situation [4,5]. and hope can encourage them to focus on the future [6]. Self-management can help chronically ill patients to feel better, as well as support from family and friends and contact with healthcare professionals [7,8]. 

Charmaz [9]. describes how people affected by chronic diseases experience social isolation and loneliness, depression, restricted activity and the feeling of being a burden. Their lifestyle needs to be changed and this is often experienced as a loss of control. To cope with the new situation, patients find ways of promoting self-management as a process of bringing order back into their lives. Everything that was experienced before the point of illness, the way life was in the past, and the hopes and dreams that were interrupted and changed, influences the experience of illness [10,11]. Röing & Sanner [1] explain how having a chronic disease can be described in terms of possessing a different living body and that this is a challenging reality.

Coping is about finding a way forward with any change of life. Lazarus & Folkman [12] define coping as a personal trait and a situational approach, and describe it as an ongoing process that changes depending on what else is happening at the time. Strategies relating to the way a person thinks and acts in order to cope with a stressful situation can be problem-focused or emotionally focused, depending on the individual’s personality [13,14]. Hoffman et al. [15], have described these strategies as a model of restorative wellbeing that reveals the interrelations between personality and affective and social cognitive variables. When the process of change shifts in the individual’s view of life in a negative or positive way it can be defined as a turning point [16].

We chose to focus on four chronic diseases: asthma-allergy, cancer, diabetes mellitus, and inflammatory rheumatic arthritis. They may seem different, but they all influence the patient’s health and the daily life situation for the rest of their life. They can all have a life-threatening component, even if the degree of threat may vary in common opinion. All the diseases influence daily life in both a practical and an emotional way. For example, patients with diabetes mellitus have to change many parts of their daily life to try to maintain good glycaemic control and monitor their food intake during the day [17]. Similarly, patients with inflammatory rheumatic arthritis change their daily life in relation to fatigue, pain, and stiffness [18]. Patients with chronic diseases experience limitations and have to develop strategies to increase their level of function in daily life [12,19]. Cancer patients have often been documented as having a traumatic disease that has developed into being chronic and life-changing but, in contrast to the other diseases, cancer can bring forth feelings of fear and uncertainty about the sufferers’ lives in the future [20]. All chronic diseases influence both physiological and psychological wellbeing, and can cause anxiety, depression and affect physical wellbeing. This can also be experienced as a loss of control that can lead to an acute crisis or reduced quality of life and can cause psychological distress and reduce emotional wellbeing [21,22]. So even if the diseases differ regarding risk factors, treatment and progression, we have focused on those aspects that can be important and similar, irrespective of other factors related to these diseases. There are also many other diseases that can result in a similar situation regarding view of life, life-changing needs and how the disease influences the patient’s life. The four diseases in this study are, however, commonly occurring in health care, we had the possibility to approach these patients and the knowledge gained may be helpful to other patients with such life-changing diseases.

This study reflects important topics that have been previously discussed in the literature and, through the use of a qualitative design, can possibly increase knowledge at a deeper level. We aimed to gain a deeper knowledge of how the participants experienced living with a chronic disease and examine how the disease can influence the participants’ daily life and thoughts. In addition, we describe the strategies that participants adopted to cope with the new situation and their experiences of being supported.

## 2. Materials and Methods

### 2.1. Study Design

A qualitative design was chosen to gain a deeper understanding of how chronically ill participants experience life-changing, social processes in their network and their related need for support. In-depth interviews were chosen to collect data to help evoke personal experiences and perspectives on the topics described in the aim of this study [23,24]. Using a content analysis shows the essential meaning of the participants’ experience. This qualitative research has a hermeneutic view with an interpretation of the participant’s experience of living with a chronic disease.

### 2.2. In-Depth Interviews

The interviews were conducted in Swedish by two of the authors, both of whom are social workers (one with a PhD), and by three social workers based at the hospital. All had worked for a long time with conversations with patients in their daily work and had extensive experience of meeting patients with serious diseases. Before the interviews took place, the specifics of research interviews were discussed with all the social workers who participated. None of the interviewers had any connection to the participants. In-depth interviews were conducted using a conversational approach. This meant that the interviews were conducted as conversations, and the participants were asked to tell their story and allowed to provide information in a variety of ways. No repeat interviews were carried out and no transcriptions were returned. 

The participants decided where the interview should take place, either at the hospital, in her/his own home or by telephone. In the face-to-face interviews, only the participant and the researcher were present. The interviews were recorded and their duration was mostly between 30 min and 1 h 30 min. No notes were taken during the interviews and they were transcribed verbatim. Any clarification of what the participants meant in certain areas was dealt with during the interview.

To ensure adherence to the research questions throughout the interviews, certain key questions were highlighted. These questions were produced from a pilot study in which other participants in the same situation were interviewed. The key questions were: Did the illness influence your life? In your experience, has the illness influenced your way of life? What has been helpful and supportive?

### 2.3. Settings

The study was conducted at the medical, rheumatology, lung, palliative and oncology clinics at a university hospital in western Sweden. We recruited participants affected by diabetes mellitus, asthma-allergies, inflammatory rheumatic arthritis, and cancer in both curative and early and late palliative stages. They were all patients at the hospital, which meant that the disease was serious and needed treatment on the open ward at the hospital. The study was approved by the Regional Ethical Review Board in Gothenburg (423-15).

### 2.4. Data Collection

Data were collected over a seven-month period between October 2015 and May 2016.

Information about the study was presented in a brochure that was available in the waiting rooms at the hospitals where the participants were treated. After being informed of the study, those who were willing to participate sent a signed consent form in a pre-paid envelope to the researcher, who telephoned the participant and arranged when and where the interview would take place. A total of 138 brochures were distributed in waiting rooms and 122 of them were taken. Of the 45 persons who gave their consent, 41 participated. The reason given by three of those who withdrew their participation when they were contacted was that their disease had become worse. One person could not be reached. When the information and invitation to participate in the study are open in the form of a brochure in a waiting room, as in this study, there is no way to send a reminder to patients. This means that only those who feel that they want to participate in an interview send in their consent.

The inclusion criteria were that the participants should have one of four diseases: asthma-allergy, cancer, diabetes mellitus or inflammatory rheumatic arthritis. The participants should be over 18 years and able to understand the Swedish language. The invitation to the study was open and only those who felt that they wanted to participate filled in their consent. All of the participants who responded were over 18 years and all understood Swedish. Consequently, no one was excluded by the researcher.

### 2.5. Data Analysis

The qualitative content analysis method goes beyond counting the frequency of certain words in the text by interpreting the latent content that emerges from the text. The analysis was performed with close examination of the topics and themes defined in the research question [25]. The content analysis was carried out with close cooperation between the authors and social workers who had conducted the interviews. The results of the interviews were discussed together to ensure that they were interpreted as objectively as possible. Saturation was found in the themes described in the result. Any discrepancies in the analysis were discussed until agreement was reached, and a common description was then formulated. In the analysis, the diagnosis groups were compared to find similarities within them and differences between them. The content analysis comprised several steps. Initially, text that was relevant to the research issues was marked. Units of meaning were then identified and grouped into codes. These codes were subsequently discussed in the whole group, compared, categorized and labelled. In the final step, the categories were structured into sub-themes and themes [26,27].

### 2.6. Trustworthiness

Many factors need to be taken into account to ensure trustworthiness and validity of qualitative research. Compared to using a questionnaire, in-depth interviews make it possible to acquire a deeper understanding of the participants’ experiences of a chronic disease. Everyone has experiences in his or her life that may influence the particular questions that are asked and the interpretation of what is said. To ensure credibility, the participants in this study were patients with a chronic disease and, despite differences in the diseases, the results revealed many similarities between them, for example how they felt and thought about their disease as well as how they found similar strategies to cope with their new life situation. Data saturation (when the phenomenon became stronger and similarities between the participants more evident, cohesive and consistent; [28]) was found, which strengthens the trustworthiness of the results and the transferability to others in the same context, i.e., other patients who are affected by a chronic disease. Trustworthiness was maintained by developing clear coding procedures/coding definitions and peer debriefing with participating interviewers. 

To reduce the influence of the researchers’ preunderstanding, and enhance and facilitate the analysis, specific probing questions were identified to clarify the narrative in relation to the research questions [29,30]. To ensure credibility, all of the researchers in this study were professionals who had worked for a long time with conversations with patients in their daily work and they were aware of the difference between these conversations and interviews and treatment. Despite differences in the participants’ age, gender, and type of disease, the results revealed many similarities supporting their generalizability [27].

## 3. Results

There were unequal numbers of participants in the different diagnosis groups. Most of the participants had cancer or diabetes mellitus. Most were female, over the age of 50 years, and well educated. Only 15 were employed and 13 were currently on sick leave. Most of the participants lived with someone. For participant characteristics, see Table 1.

When the interviews were analyzed, we found three common themes described by the patients. We also found some sub-themes that described the themes in more depth and the potential consequences of the disease. The three overall themes were: Life-changing, Strategies, and Support. There were many similarities and some differences between the different diagnoses. The differences occurred mostly in the life-changing theme. See Table 2.

### 3.1. Life-Changing

The participants often came to understand their new life situation when they realized that the disease was a turning point in their life. This could happen when they were given the diagnosis, when they were informed about the treatment and the prognosis, or when given information about the progression of the disease. At this point, the participants realized that this was a change that would last for a very long time or forever. 

The theme *Life-changing* consists of four sub-themes that could explain how life had been changed when they had the diagnosis of their disease. The four sub-themes were *The turning point*, *Thoughts of death*, *Daily life changes* and *Change of identity*.

#### 3.1.1. The Turning Point

The turning point can be seen as a negative experience if the disease is perceived to be a threat to the patient’s life, or as a more positive experience if it is perceived that living with the disease and making changes to lifestyle will be possible. However, it can be a negative experience in relation to all of the changes the participants may have to make, even if the disease is not perceived as life-threatening.

The participants who were affected by rheumatic conditions and by asthma-allergy had often experienced a long period of illness before receiving their diagnosis.

“For me it was only positive to have a diagnosis when things were as they were.” (Person affected by asthma-allergy)

Participants affected by cancer and diabetes often felt that they received their diagnosis after an acute event. The turning point for these participants could be perceived in a positive way, when they found a way of coping with the disease, or a negative way, when they realized that this could worsen their quality of life.

“It was really a relief when at last I was told what it was. Then I could understand why I behaved the way I did.” (Person affected by diabetes)

“Then I understood that this was something that just could not be treated and then disappear, instead it was more serious.” (Person affected by cancer)

#### 3.1.2. Thoughts of Death

In this study, many of the participants felt that the disease, irrespective of which one, was a threat to their life. Thoughts of death were more common in those affected by cancer but were also reported by those affected by diabetes and asthma-allergy conditions and, to a lesser extent, by those affected by inflammatory rheumatic arthritis.

“I am quite open with myself … death will come earlier in my life than if I had not had this disease.” (Person affected by diabetes)

“I immediately saw myself as dead … and I wondered when the funeral would be.” (Person affected by cancer)

#### 3.1.3. Daily Life Changes

The new life situation could mean that everyday life had to change. For the participants affected by diabetes, the change was obvious because they had to, for example, check their blood glucose level every day and take control of what they ate and when.

“And then the blood glucose level has to be checked, sometimes 5–6 times during the day and that is not so easy to do. … So I believe I am being held captive by that.” (Person affected by diabetes)

For participants affected by cancer, for example, treatments were difficult and there were many visits to the doctor. This was something they had to prioritize.

“Having all this treatment, it is so laborious.” (Person affected by cancer)

The participants affected by asthma-allergy had to consider their surroundings so as not to expose themselves to precipitating factors.

“I cannot visit my friends if, for example, they have a cold; so I am isolated.” (Person affected by asthma-allergy)

For the participants affected by rheumatoid arthritis, the pain influenced their lives.

“So much is affected, not just that you are in pain more or less all the time.” (Person affected by inflammatory rheumatic arthritis)

#### 3.1.4. Change of Identity

Another life-changing theme was the change of identity. This was perceived by the participants as questioning how they thought of and saw themselves, as becoming another person with a patient identity.

“I have become a cancer patient instead of XX [own name].” (Person affected by cancer)

“Both my self-esteem and my self-confidence were affected and it was like an identity crisis … who am I now?” (Person affected by inflammatory rheumatic arthritis)

They were no longer independent; they had become patients who depended on others. Having an identity as a patient signified that they were no longer experts in their own self; someone else now possessed the expertise in relation to the disease, and they must now depend on someone else’s knowledge and trust this person to help them. This experience brought about feelings of having less trust in themselves. It was common among the participants affected by asthma-allergy and diabetes to feel some level of guilt because they had to make demands on their friends, such as where to meet. They could not eat all the food that they might be offered, and they saw themselves as difficult to socialize with.

“They don’t want to hang out with me because they can’t bring their dogs … and if I am invited to a party I feel sad that the dogs can’t be around.” (Person affected by asthma-allergy)

“I am not able to take the initiative, instead I want others to do that.” (Person affected by diabetes)

### 3.2. Strategies

The next theme was finding strategies to cope with the new situation. Strategies can provide a way to handle difficult situations. They can include both practical arrangements and ways of thinking about the situation. Finding their own strategies to cope with the new situation often started with the participants’ first reaction to their diagnosis. This also depended on how the disease had started and how long it had taken for them to be given a diagnosis. It could be experienced as a relief to know what disease they had. However, it could also be accompanied by thoughts of what the future would mean in their life.

Sometime later after receiving the diagnosis, a process of creative thinking could take place and the participants could find strategies to deal with the problem. This meant that the participants would seek out strategies that suited their individual personalities. These strategies were designed to create ways to cope with their daily tasks as well as allow them to experience a good quality of life, irrespective of their disease.

The results revealed three sub-themes within the strategies that the participants had developed: *Hope for a good life, Taking control of the consequences of the disease for their body and in daily life and Finding meaning in the new situation and a life worth living*.

#### 3.2.1. Hope for A Good Life

Hope was a strategy that provided opportunities to be able to look forward to the future. If you do not have hope you cannot see a future. It could be receiving better treatment or surviving. Having hope helped them to live in the present.

“This will be a life filled by worry and uncertainty for some years to come … I have a little more hope of surviving now, and every time I come for a check-up and it looks good, the chance of that increases.” (Person affected by cancer)

“Acceptance and hope that they find some new medications and treatment.” (Person affected by asthma-allergy)

“You live in hope all the time… that this must help now.” (Person affected by inflammatory rheumatic arthritis)

#### 3.2.2. Taking Control of the Consequences of the Disease for Their Body and in Daily Life

This sub-theme relates to taking responsibility for the disease and finding ways of coping with the new situation. This was about patients taking control of the treatment as well as what needs to change in their life situation to feel well. If they take control of the outcome of the disease in their daily life, it means that they can have the possibility to choose how to deal with the problems that the disease raises in daily life. This gives a feeling of being able to choose how life will be in the future, whether in a good or a bad way, despite the disease.

“I wasn’t shocked by it, instead I thought that I have to take it as it is and make the best of it.” (Person affected by cancer)

“Not thinking that this means the end when you are seriously ill but doing something positive about it.” (Person affected by a form of inflammatory rheumatic arthritis)

“When I was diagnosed it was a relief, yes, now I can do something about it.” (Person affected by diabetes)

#### 3.2.3. Finding Meaning in the New Situation and A Life Worth Living

This sub-theme was about finding a way of living a good life, despite the disease. It was about seeing and finding that the change in life was not the end of life but a possibility for finding other ways of living and that other values can arise in life. What they had earlier regarded as the meaning of life could be changed and it was necessary to find a new meaning of life in order to feel that life was worth living. The meaning could be both practical as well as the new life situation and how they will live life.

“It is important to find your own way to live and to function.” (Person affected by diabetes)

“Today, I have accepted things as they are, and I can enjoy life more.” (Person affected by asthma-allergy)

“I can enjoy small things, such as just being able to sit in the sun, and this is very important to me.” (Person affected by inflammatory rheumatic arthritis)

### 3.3. Support

The last theme was support, including both practical and emotional, to help the participant to cope with the new life situation. Support came mostly from the private network and was seen as important. Support from health care was also requested and could take many different forms. The sub-themes were *Support from the private network* and *Support from healthcare professionals including accessibility and information*.

#### 3.3.1. Support from the Private Network

Support was obtained from both the participants’ private network of family and friends, and from work colleagues. The private network was reported as most important for all of the participants, regardless of their diagnosis.

“I have very strong support from my children. They are absolutely fantastic in this situation … and also from my lady friends who offer to help me with shopping and cleaning … but I find it difficult to accept support, but I know I can get support when I really need it.” (Person affected by inflammatory rheumatic arthritis)

“I have met new friends who I can talk to through the hospital.” (Person affected by asthma-allergy)

Support could come in the form of talking and reflect about the new situation, getting practical help from friends and colleagues by helping them to understand the new life situation, or practical help to be able to continue working in this new situation. Talking to others could be experienced both as a positive or a negative feeling, depending on their own view of the situation.

“In the beginning I didn’t want to talk to anyone about this. No … I only talked to those really close … but when I knew more about the disease I could talk to more people.” (Person affected by cancer)

“For me it is easy to talk about my illness to my loved ones. I am not ashamed.” (Person affected by diabetes)

#### 3.3.2. Support from Health Care

Support could also be provided by the healthcare service, in both the routines, information and the follow-up system.

Support was described as being drawn from the empathy that the healthcare staff showed the patients; with them being seen as a person not only as a patient.

“To have a relationship with the healthcare staff that works well, that is important.” (Person affected by diabetes)

“To have the same nurses and assistant nurses, familiar faces make you feel like a human being … it means so much.” (Person affected by inflammatory rheumatic arthritis)

Accessibility was also important; it allowed the participants to contact the healthcare staff when they needed to. This was a high priority for the participants and accessibility meant both having access urgently and having a professional contact who knew their problems. Continuity follow-up was important to the participants and contributed to the feeling that they were being cared for and gave the possibility of developing a relationship with the staff.

“These physicians … that they are available … that I always get feedback within a couple of days. It has been an enormous support.” (Person affected by asthma-allergy)

“I have my medical supervision at the hospital and that makes me happy … They check on me quite often and I can call if I have any questions.” (Person affected by cancer)

Information was also important to the participants. Information could be provided in individual conversations and in direct contact with the healthcare staff. The most important way to receive information was verbally, and the participants described how it should be individualized and often required repeating.

“I have to have the possibility to ask questions.” (Person affected by asthma-allergy)

Written information was also needed, both for themselves, but also for their own network and was a complement to the verbal information and a resource for later. Some of the participants described how they had to search for the written information themselves, and this was often regarded as a negative experience. It was more helpful if they were given tips on where to find reliable information.

“I have had lots of information about all the steps and many books and brochures … I don’t read them; it is my mother who has read them and then she tells me. But it is good to have them at home.” (Person affected by cancer)

“It was not only verbal information that was needed because you can’t remember what the doctor said. (Person affected by diabetes)

## 4. Discussion

This study focused on how a chronic disease can be experienced and can influence a patient’s life and how she/he might cope with the new life situation. This experience and influence was identified by all of the participants in our study, across the four diseases, and the findings have shown both similarities and differences between the diseases. Van Houtum et al. [7]. found that it is the patient’s own perceptions of her/his illness as well as the actual changes that determine how the patient experiences the disease. Öhman et al. [10]. described how severely ill patients hover between suffering and enduring, an experience that was also described by the participants in this study. This was identified in the analysis of the interviews in the themes *Life-changing, Strategies*, and *Support*.

According to Kristjansdottir et al. [8], personal strengths, consisting of external and internal strengths as well as self-management strategies, can be helpful for people with a chronic disease. In the process of finding a new life-situation, all three themes were common and affected each other.

### 4.1. The Life-Changing Theme

In the life-changing process, the possibility to receive information about their condition and their identity as a patient were described as important. The information was commonly described as the first time the participants realized that it was a disease that they would have for the rest of their life and one that would influence their daily life in different ways. This could be experienced as a turning point, a change in their life. A turning point can consist of learning, as described by Berglund [31], containing new thoughts, feelings and actions as well reflection on experiences. Rise et al. [32], describe a turning point as positive and could include having to absorb new information, take responsibility for one’s own health and make changes in one’s life, which has also been seen by others [33]. However, it could also be felt as a negative turning point in that, from this point onwards, their quality of life would worsen and the condition would be a hindrance in their life [34].

In our study, this was described by participants affected by cancer, both the diagnosis as an experience of having a fatal disease with the fear of death and the treatment of that disease. A surprising finding in this study was that the existential view of death was a fear expressed by the participants affected by all of the diseases. The existential thoughts of having a fatal disease did not exist only for the participants affected by cancer, but also for those diagnosed with diabetes and allergy-asthma and, to a lesser extent, participants affected by inflammatory rheumatic arthritis. We have not found this previously described in the literature, other than in relation to cancer, organ failure, and frailty in older patients [35]. Innes & Payne [36] found that patients often want to have some ambiguity about their future in order to strengthen the hope of being a survivor of the cancer disease, something that was also identified in this study.

Another part of the life-changing disease was that the participants could perceive their identity as changed, since now they identify themselves as persons with a chronic disease and think of themselves as patients. Being a patient seems to be connected with being dependent on medication, visits to healthcare services, having different forms of treatment, and so on. This includes others’ reactions to them becoming sick persons as well as their own way of perceiving their self-identity as being either sick or healthy [1]. The results of a study of rheumatology patients may apply, in a broader sense, to the identities of all patients affected by such conditions in that they recognize themselves as both ill and healthy at the same time, with the statement “I am not only a disease, I am so much more” [37]. The patient can also be an expert in her/his own disease and treatment as shown by Sanderson & Angouri [38], concerning patients with rheumatoid arthritis. Being a survivor can influence identity, particularly for cancer patients, for whom fear of death is common. The self-identity as a cancer survivor is an identity that may be embraced after time when the disease has been treated for longer [39].

### 4.2. Strategies

According to the new life situation, the participants were forced to find strategies to cope with the new situation and daily life, described in the second theme. Coping has been defined as “constantly changing cognitive and behavioural efforts to manage external and/or internal demands that are appraised as taxing or exceeding the resources of the person,” according to Lazarus & Folkman [12] and Folkman & Tedlie Moskowitz [13].

One of the sub-themes that emerged from the strategies in this study was the hope that participants developed to cope with the new situation. In relation to other strategies, maintaining a sense of hope was identified as a central need for all participants, irrespective of the disease. Duggleby et al. [6] have formulated an explanation of the characteristics of hope as being a process of moving between positive reappraisal and transcendence, and that this is a process that persons with a chronic disease must go through.

In this study, the participants described how their basis for developing a positive coping strategy started with their reaction to the way their new life was affecting them. This was experienced as a turning point and when a new way of living became obvious. From this, they described how, when given the diagnosis, this was perceived already as a starting point for developing either positive or negative coping strategies. A positive way of coping occurs when a person experiences a life-changing disease and feels that she/he has control over how the disease develops. This can be experienced as modifying their levels of activity to suit their current functional ability, and developing an awareness of their functional limitations [40]. Learning more about the effect of the disease and what might worsen the symptoms [41] helped them to accept the situation in their mind and realize that their disease would influence their future life-situation. Coping could also be through planning for daily activities and using assistive devices [42]. Finding appropriate strategies is an ongoing process related to what happens with the progression of the disease, its treatment and the influence that it has on the patient’s daily life. The strategies were used in both an emotional and a behavioural way to be helpful. Self-management groups could be helpful in find strategies but also as a support for the life-changing process [41,43] and being able to find resilience in their life situation.

### 4.3. Support

Support could consist of both informal support from their own private network as well as formal support from professionals. Many participants pointed out how their own social networks were a strong source of support. This was provided in the form of daily support which helped them to find new ways of living connected to both acceptance of a new life situation as well as finding new strategies to cope with this new situation [44,45]. The support consisted of both practical and emotional support to help them cope with the feelings that arose from focusing on the disease [14,46]. When experiencing a life-changing disease, the support received from family members could include other dimensions, as described and identified in this study. This has been reported earlier as a form of *shared respite*, as a way of finding new energy in that they are allowed to shelter in a secure environment with their family, *a place of relief* where everyone knows what life is like living with these conditions [47]. It could also be that they can reflect with other family members, and this can be perceived as a form of *reflection and re-creation*, where the family can try to find new ways of living in daily life and explore elements of their new collective identity [47]. All the participants commonly experienced that the professional staff played an important role in supporting them, both practically and in psychological and emotional matters. Emotional support given by the professionals should be offered by the clinical staff who are familiar with the particular problems that can arise. There was also a strong wish from all the participants to be given both verbal and written information; it can be a challenge for healthcare professionals to create routines for providing this information. This can be especially important for patients who have a disease that will influence them for the rest of their lives.

Information about the disease they were suffering from was perceived as being important and was always used as grounds for determining how to understand the situation [4]. In our study, there was a strong wish for information, both verbal and written as a complement, and this was needed during the whole period of illness.

For all participants, irrespective of diagnosis, the information focused on the treatment of the disease, the medication to avoid symptoms, how the disease might influence daily life, and what the support network might include. The different ways of providing information must always be tailored to the patient’s own needs, as not all forms of information will suit all persons. The participants affected by diabetes needed to have control over the food they ate and to measure their blood sugar levels, and the literature describes the various ways of providing information about diet and self-management to this particular group. This could take the form of, for example, participating in group intervention and patient education programmes [7,43]. There are also interventions given as web-based programmes [48]. The participants affected by asthma and allergy needed information to identify all of the things in their food or the environment that might bring about the symptoms that affected their body and breathing. A recent study found that smart-phone-based applications could perform surveillance and track trending air pollution levels which could potentially provide important information to participants with asthma [49]. Being provided with appropriate information can increase quality of life, according to Bäurle et al. [41]. In our study, the participants affected by asthma and allergy had difficulties meeting with other persons who did not understand the severity of being exposed to allergens in food or in the environment, or of the severity of the symptoms of the disease. This created limitations, both in their activities and in relation to food. Sadatsafavi et al. [50] described how such limitations on everyday activities were correlated to quality of life. Some participants in our study felt that they experienced many losses in their daily lives and in their relationships with others, and that the disease really worsened their quality of life. The hindrance that the disease causes can milder the disease outcome with information [41,51].

For the participants in this study affected by inflammatory rheumatic arthritis, the life-changing effect was that pain was constantly present and could have a strong influence on the activities in their daily life. Ahlstrand et al. [40] have also reported these restrictions and that pain was related to fatigue, stress, and mood. The negative outcome in life after being diagnosed with rheumatic arthritis has also been described by Lööf et al. [52]. The information of how new medication and non-pharmacological therapies can also decrease the burden caused by inflammatory rheumatic arthritis [53].

### 4.4. Study Limitations

Qualitative studies often have a small sample size, which may limit generalizability, but in this study, the sample was quite large. The participants gave detailed information that provided insight and they felt that they could talk about their disease from a research point of view. The participants were able to express their feelings and, despite there being four diseases in this study, the results pointed to similarities between them. Patients who do not want to talk about their disease may have a different experience, but in this study, we were able to identify an overview of how such a disease can influence daily life. Many of the participants were women, which could influence the result from a gender point of view. Most of the participants were highly educated and this might have influenced the result, for example, by them being able to express their stories verbally and give suggestions for improvements in health care. Another limitation could be that all participants speak Swedish and those who do not speak the language may have given another perspective of the research question. Future directions for research about chronic diseases could be through a quantitative study to evaluate a larger number of patients with chronic diseases.

## 5. Conclusions

A conclusion from the results of this study is that chronic illness is part of life, both from a physical and a psychosocial point of view, whether it is through chronic pain or a disease that influences physical health and functions, autonomy, changing in the daily life situations, freedom, and identity. The new in this study was that the fear of death is common in more than one of the diseases. It was obvious that persons suffering from a chronic disease need to challenge the meaning of their life and change it in the practical life situation. Therefore healthcare professionals need more knowledge of what the process of coping with such life-changing matters may be and what could strengthen a patient with a chronic disease in order to give them a sense of empowerment in their lives. Having increased knowledge of patients’ experiences can, therefore, be the foundation on which healthcare professionals can develop special teams with or models for new methods to provide support in the healthcare system, something that the informal network may not be able to give. This can, furthermore, help the patients to cope with the new situation and to meet the questions of both physical, psychosocial and existential dimensions that this process can include.

## Figures and Tables

**Table 1 diseases-07-00053-t001:** Characteristics of study participants.

	n
**Gender**	
Male	6
Female	35
**Age years**	
20–29	3
30–39	7
40–49	6
50–59	9
60–69	8
70-	8
**Employment**	
Employed	15
Sick leave	8
Retired	11
Employed and sick leave	5
Employed and retired	1
No answer	1
**Education**	
Primary school	2
High school	9
University	29
No answer	1
**Living situation**	
Living alone	14
Cohabiting	26
No answer	1
**Disease**	
Asthma–allergy	5
Cancer	18
Diabetes mellitus	11
Inflammatory rheumatic arthritis	7
**Years with the disease**	
Less than 1 year	2
1–2 years	12
2–4 years	2
4–6 years	4
>6 years	21

Total, *n* = 41.

**Table 2 diseases-07-00053-t002:** Similarities and differences.

	Asthma-Allergy	Diabetes Mellitus	Cancer	Inflammatory Rheumatic Arthritis
**Life-Changing**				
The Turning point after having the diagnos	After a long period	After a short period	After a short period	After a longer period
Thought of death	Not common	Common	Common	Not common
The daily life changing	Be aware of environment for allergens	Blood sugar control every day	Especially during the treatment period	The pain influence the daily life
Change of identity	Feeling guilty about causing problems for their friends	Feeling guilty about causing problems for their friends	Have an identity as a patient	Have an identity as a patient
**Strategies**				
Hope for a good life	Similar	Similar	Similar	Similar
Taking control of the consequences of the disease for their body and in daily life	Similar	Similar	Similar	Similar
Finding meaning in the new situation and a life worth living	Similar	Similar	Similar	Similar
**Support**				
Private network, talking and practical help	Similar	Similar	Similar	Similar
Health care professional. Information and accessibility	Similar	Similar	Similar	Similar

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
