# Peer review of "“Not Thinking that This Means the End When You Are Seriously Ill but Doing Something Positive about It”—A Qualitative Study of Living with A Life-Changing Disease"

_diseases, 2019, doi:10.3390/diseases7030053_

Round 1
Reviewer 1 Report
Thank you for inviting me to review the paper on “Not Thinking That This Means the End When You are Seriously Ill but Doing Something Positive about It” – a Qualitative Study of Living with a Life-Changing Disease”. This is an interesting quality study and deserves publication. I have the following recommendation:
1. Line 69: The authors said: “All chronic diseases influences both the physiological and the psychological well-being and can cause anxiety and affect physical well-being.” I recommend the authors to include depression and related references. Physical and physiological well-being is redundant. Please modify the references as follows:
All chronic diseases influence both the physical and the psychological well-being and can cause anxiety and depression (Lu et al 2012, Vu et al 2018, Khue et al 2019, Ho et al 2011)
Reference for Asthma-allergy, depression and anxiety
Lu Y et al Prevalence of anxiety and depressive symptoms in adolescents with asthma: a meta-analysis and meta-regression. Pediatr Allergy Immunol. 2012 Dec;23(8):707-15. doi: 10.1111/pai.12000. Epub 2012 Sep 9. PMID:22957535
Reference for diabetes and depression:
Vu HTT et al Depressive symptoms among elderly diabetic patients in Vietnam. Diabetes Metab Syndr Obes. 2018 Oct 23;11:659-665. doi: 10.2147/DMSO.S179071. eCollection 2018. PMID:30425543
Reference for cancer, depression and anxiety
Khue PM et al Depression and Anxiety as Key Factors Associated With Quality of Life Among Lung CancerPatients in Hai Phong, Vietnam.
Front Psychiatry. 2019 May 16;10:352. doi: 10.3389/fpsyt.2019.00352. eCollection 2019. PMID:31156487
Reference for rheumatoid arthritis, anxiety and depression:
Ho RC et al Clinical and psychosocial factors associated with depression and anxiety in Singaporean patients with rheumatoid arthritis. Int J Rheum Dis. 2011 Feb;14(1):37-47. doi: 10.1111/j.1756-185X.2010.01591.x. Epub 2011 Jan 24. PMID: 21303480
2. Line 453: the authors stated that “The participants affected by asthma and allergy needed information to identify all of the things in the food or the environment that might bring about the symptoms that affected their body and breathing.”. This statement needs more clarification. Asthma is affected by air quality and the author should provide an example. Please add the following statement:
…. their body and breathing. A recent study found that smartphone-based application could perform surveillance and trending of air pollution level which could potentially provide important information to participants with asthma (Zhang et al 2014).
Reference
Zhang MW et al Methodology of developing a smartphone application for crisis research and its clinical application. Technol Health Care. 2014;22(4):547-59. doi: 10.3233/THC-140819. PMID:24898865
Author Response
Here comes our comments and changes to the reviewers valuable comments to the article. The comments has been marked with bold text. In the manuscript changes and added text has been marked with yellow colour, red for text that shall be removed. Thank you for the valuable references you have given.
Reviewer: 1
Number 1
We have now add a reference and add that depression can be caused by the chronic disease
Number 2
Thank you for the statement. We have now add this in the text and add the reference too
Reviewer 2 Report
Comments to the Authors:
The authors, Inger Benkel. et al., report a Qualitative Study of Living with a Life Changing Disease.
They conclude that patients have to make changes in their daily life, and these can bring different feelings and restrict activity. Health care professionals need more knowledge of what the process of coping with such life-changing matters and what could strengthen to give a sense of empowerment in their lives.
In characteristics of study participants, there are excessive female, and they refer about chronic diseases, they don’t survey about hypertention and cerebral infarction.
Furthermore, statistical analysis is not used and the result don’ t have novel statistical significance.
Table submitted is only one table and no figure miscellaneous.
From this point of view, the conclusion could not be guided.
The authors have to emphasize the novelty of this report.
Author Response
Reviewer: 2
It is correct that we not refer to other than those four chronic disease we choose. We have state why we choose those and the result can be transferable to other chronic diseases when they have similar outcome as those diseases.
We did a qualitative design with in-depth interviews and therefore we do not use any statistical analysis.
In the qualitative analysis it is not common to use figures or statistical data. The table for the participant’s characteristics is a way of given an overview of the participants which can be easier to see than describe in words
Out of the qualitative research methods the conclusion is from that point of view
We have add some words to emphasize the novelty the study found.
Reviewer 3 Report
The Authors of this Swedish study interviewed 41 adult patients to explore the influence of four chronic diseases [asthma-allergy, cancer, DM and rheumatic disorders] on daily life, coping strategies and experiences of support.
Although the findings showed both similarities and differences between the diseases, it was concluded that -overall- the influence regards changes in daily life, along with a series of feelings and activity restrictions. The Authors suggest that the knowledge of their study's information is important to health care professionals in order to be prepared to strengthen patients with a sense of empowerment in their lives.
The Ms is well written and easy to understand.
There are a few issues which need attention.
1. A merit of the study is to have grouped together more conditions. Atltough this is a qualitative study, this referee suggests that - thanks to the study design- the Authors make additional efforts in order to prepare and include a few Tables summarizing synoptically the main similarities/differences found in patients' answers. This may help the reader to better distinguish generic from disease specific support needs of people with chronic illness.
2. A better distinction between need of Social support and Health care professionals support should be done. Examples of issues to be discussed: patient education programs to promote self management; emotional coping with a condition requiring an ongoing support from community and clinical resources; need for the creation of Teamwork and Collaboration model with a shared meaning of teamwork and interprofessional collaboration among health care professionals also in primary health care settings, etc-
3. Authors should indicate in the MS if they have information on patients' pharmacological treatment with antidepressant drugs
4. Recent studies underline chronic ilnesses patients' ability of acquiring resilience, in a dynamic process of learning in response to new challenges by means of a combination of behavioral and emotion management strategies (Shaw et al Arthritis Care Res 2019). Have the Authors explored this aspect in their interviews ? In any case, they should discuss that the awareness of these strategies may benefit patients, healthcare providers and researchers developing behavioral interventions and social support programs in the context of chronic diseases
5. Authors should mention the existence of predictors of Acceptance of Life with the Disease, which include coping strategies, HRQoL, and General Self-efficacy (Wytrychiewicz et al. Psychol Health Med. 2019). Psychological factors play an undoubted role in adapting to life with chronic illness. Self-efficacy is a personal resource which psychological/medical interventions should focus on
Author Response
Reviewer: 3
Number 1
We have now done a Table with similarities and differences.
Number 2
We have add some sentence in the Discussion and Conclusion sections for a better distinction between social support and health care professionals support in the text
Number 3
We have no information if the patients is treated with antidepressive drugs. This was not anything that were commenting in the interviews and not a question which were asked specific about.
Number 4
We have not explore this but as you point out this is an outcome that may more be experienced and helpful in the patient’s new life situation. We have add such a sentence in the Discussion section.
Number 5
We had a part of accept the situation connected to the strategies they use as you pointed out. We have clarified that now in the text
Round 2
Reviewer 2 Report
A whole report is well written. Their viewpoint of both physical and psychosocial is highly original.
Data they presented as figures are scientifically sound.
Author Response
Answers to reviewer 2;
Thank you for your comments. We have now checked Introduction, research design, methods, result and conclusion and improved and it with clarifications. We have also checked the language. Our new changes is marked in green color.
Reviewer 3 Report
The Authors answered adequately to this Reviewer's comments.
The MS requires some minor English language polishing/editing of Table 2 added in the revised version.
Author Response
Answers to reviewer 3;
Thank you for your comments. We have now checked the language and editing Table 2.
Our new changes is marked in green color.